# Atomic Layer Deposition of Cobalt Catalyst for Fischer–Tropsch Synthesis in Silicon Microchannel Microreactor

**DOI:** 10.3390/nano12142425

**Published:** 2022-07-15

**Authors:** Nafeezuddin Mohammad, Shyam Aravamudhan, Debasish Kuila

**Affiliations:** 1Department of Nanoengineering, Joint School of Nanoscience and Nanoengineering, Greensboro, NC 27401, USA; nmohammad@aggies.ncat.edu (N.M.); saravamu@ncat.edu (S.A.); 2Department of Chemistry, North Carolina A&T State University, Greensboro, NC 27411, USA

**Keywords:** silicon microchannel microreactor, atomic layer deposition, Fischer-Tropsch synthesis, cobalt nano-film

## Abstract

In recent years, rising environmental concerns have led to the focus on some of the innovative alternative technologies to produce clean burning fuels. Fischer–Tropsch (FT) synthesis is one of the alternative chemical processes to produce synthetic fuels, which has a current research focus on reactor and catalyst improvements. In this work, a cobalt nanofilm (~4.5 nm), deposited by the atomic layer deposition (ALD) technique in a silicon microchannel microreactor (2.4 cm long × 50 µm wide × 100 µm deep), was used as a catalyst for atmospheric Fischer–Tropsch (FT) synthesis. The catalyst film was characterized by XPS, TEM-EDX, and AFM studies. The data from AFM and TEM clearly showed the presence of polygranular cobalt species on the silicon wafer. The XPS studies of as-deposited and reduced cobalt nanofilm in silicon microchannels showed a shift on the binding energies of Co 2p spin splits and confirmed the presence of cobalt in the Co^0^ chemical state for FT synthesis. The FT studies using the microchannel microreactor were carried out at two different temperatures, 240 °C and 220 °C, with a syngas (H_2_:CO) molar ratio of 2:1. The highest CO conversion of 74% was observed at 220 °C with the distribution of C_1_–C_4_ hydrocarbons. The results showed no significant selectivity towards butane at the higher temperature, 240 °C. The deactivation studies were performed at 220 °C for 60 h. The catalyst exhibited long-term stability, with only ~13% drop in the CO conversion at the end of 60 h. The deactivated cobalt film in the microchannels was investigated by XPS, showing a weak carbon peak in the XPS spectra.

## 1. Introduction

Fischer–Tropsch (FT) synthesis involves the conversion of synthesis gas, a mixture of carbon monoxide and hydrogen, into clean synthetic petroleum fuels (synfuels). In recent years, the ability to produce clean fuels from syngas has had tremendous commercial and research values. The feedstock is typically any carbon source, such as coal (coal-to-liquid, CTL), biomass (biomass-to-liquid, BTL), and natural gas (gas-to-liquid, GTL) [1]. In general, FT synthesis is carried out using iron- or cobalt-based catalysts, with or without promoters, incorporated into high surface area supports [2,3,4]. The operating conditions vary for the type of catalyst and feedstock [5]. This thermochemical process is highly exothermic in nature, with a reaction enthalpy of −150 kJ per mole of CO conversion. Generally, three types of reactors—the fixed bed, fluid bed, and slurry bubble columns—have been extensively investigated and used on a lab-scale, the pilot-scale, and commercial scale for FT synthesis [6]. The major issues on mass transfer and heat transfer between the reactants, i.e., synthesis gas with the active sites of a solid catalyst, play a vital role in the process intensification of such exothermic reactions [7,8]. The hotspots and insufficient surface areas for reactions in the conventional reactors hinder the overall efficiency and process intensification for FT synthesis. However, the enhancement of mass and heat transfers can be achieved using miniaturized reaction zone systems, called microchannel microreactors, or microreactors for short, with large reaction surface areas for efficient FT synthesis [9,10,11]. More specifically, they can greatly enhance the surface area of the reaction zone when a large number of microchannels are considered. The higher surface areas of the microchannels can efficiently dissipate heat and enhance mass transfer for temperature-sensitive reactions such as FT synthesis. Thus, microreactors can maintain isothermal conditions throughout the reaction process and can play a significant role in the process intensification of FT synthesis and in the process economics of an FT chemical plant, such as, for instance, Pearl GTL (gas-to-liquid), which is one of the commercial FT chemical plants commissioned by Qatar Petroleum and Shell (delivering 140,000 barrels per day (BPD) is considered as the economic scale of the GTL process) [12]. Advancements in microreactor technology could therefore considerably contribute to small-scale portable and economic GTL modular chemical plants in the future. 

More recently, the use of microreactors to screen Fischer–Tropsch catalysts has received significant attention due to many advantages, such as efficient heat and mass transfer, easy to scale up by numbering or stacking [13] them together, and process safety when compared to conventional reactors [4,14,15,16]. In our previous Fischer–Tropsch studies, we used silicon microreactors, coated with silica and alumina supports containing catalysts prepared by sol-gel techniques [17,18,19]. The addition of a promoter such as ruthenium metal to FeCo-based catalysts helped to obtain a higher conversion of syngas and better selectivity towards higher alkanes [17]. Furthermore, our previous studies showed that the type of metal incorporated into silica- and titania-supported sol-gel catalysts in silicon microchannels has a significant effect on Fischer–Tropsch activity [14,18]. Interestingly, in all these studies, it was observed that the actual metal loadings were lower than the intended metal loadings in the microchannels. In addition, the sol-gel-coated catalysts exhibited very low surface areas, with a non-uniform metal distribution that led to poor interaction with reactants during Fischer–Tropsch synthesis, thereby rapidly deactivating the catalyst.

In order to improve adhesion in the microchannels, we previously developed techniques such as a modified infiltration method that takes several hours to days to coat the channels. In addition, the sealing of a Si-microreactor with glass by anodic bonding is quite challenging due to surface contamination by the catalyst slurry. Zhao et al. deposited platinum, typically less than 10 nm, by the physical vapor deposition technique (PVD) inside the microchannels before anodic bonding and conducted experiments to convert cyclohexene to cyclohexane and benzene [20]. However, since these microchannels are 100 µm deep and have an aspect ratio of two or higher, it is difficult to achieve conformal metal deposition by PVD techniques. Similarly, coating of the microchannels with sol-gel-supported metal catalysts in the microreactor for FT synthesis also lacks selectivity towards preferred hydrocarbons. This non-uniformity of active sites creates a barrier during FT synthesis and hinders catalyst performance. In contrast, atomic layer deposition (ALD) is a well-known method for conformal deposition of thin films in high aspect ratio trenches without losing film quality [21]. This technique is well-suited for the deposition of catalysts inside microchannels controlling the metal/metal oxide sites to atomic levels that maintain and improve the catalytic activity in the reaction zone. 

Herein, we report cobalt oxide as a catalyst film that was conformally deposited inside the Si-microchannels before anodic bonding for active use of the surface area of the reaction zone, providing a large number of active sites for FT synthesis. The physicochemical properties of the cobalt-based ALD catalyst were characterized using XPS, TEM, and AFM techniques. Since it is difficult to characterize the catalyst deposited in the microchannels, the thin film was investigated on the silicon wafer that was used for the fabrication of the microchannel microreactor.

## 2. Experimentation

### 2.1. Materials

Bis(N-t-Butyl-N-ethylpropanimidamidato) cobalt (II) 98%, obtained from Strem chemicals, was used as a cobalt precursor with distilled water for ALD deposition.

### 2.2. Fabrication of Silicon Microreactors

The entire fabrication process of silicon microchannel microreactors used in this study was discussed in detail elsewhere [17,22,23]. The design and fabrication of the microreactor were based on the previous research performed at Louisiana Tech University, with modifications in the reaction zone such as longer channels and the dry etching process. In order to fabricate any microdevice on a silicon wafer using conventional microfabrication techniques, a photomask is first designed using computer-aided design software. The microreactors were fabricated on a 4-inch silicon wafer with a thickness of 500 µm. The photomask and the three microreactors fabricated on the silicon wafer before anodic bonding are shown in Figure 1a. As Figure 1 indicates, each wafer consisted of three microreactors with 118 straight channels; each channel was 2.4 cm long, 50 µm wide, and 100 µm deep, designed in parallel. The design was based on the splitting of reactants and recombination of products, with catalysts deposited in the microchannels (as a reaction zone) between them.

In short, the fabrication of a microreactor involves several steps; first, a thin 10 nm titanium is deposited by e-beam physical vapor deposition, followed by a 100 nm thin aluminum film by the same method. The wafer is spin-coated with SPR 220 3.0 positive photoresist and soft baked at 115 °C for 90 s. The pattern on the photomask is transferred using the UV lithography technique and is developed in Microposit ^®^ MF^®^ 319 photoresist developer for 2 min. To make a metal mask of the microreactor design on the silicon wafer, the aluminum and titanium are wet etched, and the wafer is prepared for dry etching with deep reactive ion etching (DRIE) equipment. The pattern on the wafer shields the rest of the silicon wafer, except when exposed to dry plasma, thereby acquiring a channel depth of 100 µm. Figure 1b shows the final anodically bonded microreactor after stripping it of photoresist and the metal deposited on the wafer; it also shows one of the three fabricated identical microreactors diced from the silicon wafer. 

### 2.3. Deposition of Cobalt Catalyst Using ALD Technique

The cobalt precursor was deposited on the surface of the microchannels of the microreactor by the atomic layer deposition (ALD) technique, maintaining the reaction chamber at 220 °C. Prior to starting the ALD experiments, the precursor was heated to 70 °C and was pulsed for 0.75 s into the reaction chamber, followed by distilled water for 0.015 s. The cobalt oxide film deposited in the microchannels is schematically shown in Figure 2. The deposition of ultrathin films by this technique relies on a series of half chemical reactions of precursors on the surface of the silicon wafer. The cobalt metal precursor was pulsed into the reaction chamber that was maintained at 220 °C, with silicon substrate placed at the center of the chamber. After reacting with the silicon substrate, the metal precursor formed a reactive ligand of the metal on the surface, which is referred to as the first half-reaction. The reactor was then purged with nitrogen to remove the unreacted metal precursor on the silicon surface. The pulsing of water vapor into the reaction chamber contributed to the second half-reaction. This series of alternative self-limiting half-reactions led to the deposition of cobalt oxide from one atomic layer to the desired thickness. 

### 2.4. Experimental Setup for Fischer–Tropsch Synthesis in Silicon Microchannel Microreactor

The FT studies were conducted in a customized LabVIEW automated experimental setup, with precise control over a wide range of operating conditions, such as temperature, syngas flow rate, etc. The syngas ratio and the flowrates of hydrogen and carbon monoxide were accurately monitored and maintained by mass flow controllers obtained from Cole Palmer (with a minimum flow rate of 0.1 sccm and a maximum flow rate of 1 sccm). Nitrogen was used as an internal standard for the reactions from the mass flow controller obtained from Aalborg (0–10 sccm). The pressure in the upstream and downstream of the process was monitored using Cole Palmer digital pressure gauges. The data were fed to the LabVIEW system to control the overall pressure in the reaction zone. The products in the outlet gas stream from the microreactor placed in the reactor block was quantified by an online GC-MS system. Figure 3 shows all the devices on the optical table used for FT synthesis in the Si-microchannel microreactor.

## 3. Characterization

The morphology of thin cobalt film on a silicon wafer was investigated by transmission electron microscopy (TEM) and atomic force microscopy (AFM). X-ray photon spectroscopy (XPS) measurements of the catalyst film were carried out using an Escalab Xi+ Thermo scientific spectrometer obtained from Thermo Scientific, West Sussex, UK. All the spectra were recorded using an aluminum (Al) anode, with the same spectral parameters for each sample. The intensity of the spectra was calculated after the baseline subtraction and deconvoluted using the gaussian method [24,25]. The spectra for the as-deposited, reduced sample, and the deactivated catalyst were recorded with similar parameters for easy comparison.

## 4. Results and Discussion

### 4.1. TEM

The deposition of cobalt nanofilm by the ALD technique on the silicon wafer was confirmed by the cross-sectional TEM image depicted in Figure 4. The image indicates the presence of cobalt oxide as a polygranular film on the silicon wafer, and it was uniform throughout the sample. The approximate depth of the cobalt film, determined by TEM, was around ~4.5 nm. The presence of cobalt was also confirmed by energy-dispersive X-ray spectroscopy (EDX) of high-magnification TEM, as shown Figure 4b,c. The EDX mapping confirmed the presence of cobalt film on the surface of the silicon wafer.

### 4.2. AFM

Figure 5 shows the 500 nm × 500 nm image of the cobalt present on the surface of the silicon wafer, recorded by AFM. The morphology of the film showed grain-like structures that were not very closely packed. It is also worth noting that these columnar grain cobalt structures had significantly less roughness when compared to powdered catalysts coated inside the silicon channels of the microreactor [13,22]. It is speculated that a bit of roughness of the film helps it to adhere to the reactants on the active sites during FT synthesis, facilitating good interaction of reactants and catalysts.

### 4.3. Chemical Oxidation States of ALD-Deposited Catalyst

The chemical oxidation state and surface composition of the cobalt species deposited in the microchannels were determined using the XPS technique. Figure 6 depicts the XPS spectra for the cobalt 2p spin split of the as-deposited cobalt oxide film on the surface of the silicon wafer. The binding energies for Co 2p_/2_ and Co 2p_1/2_ were centered at 780.93 eV and 796.4 eV, respectively. They confirmed the presence of cobalt in the Co^2+^ and Co^3+^ oxidation states. The shake-up peaks, also known as satellite peaks, were observed at 788 eV and 803 eV, respectively. The shake-up peaks for Co2p_3/2_ and Co 2p_1/2_ on the cobalt deposited on the silicon wafer also indicated the existence of paramagnetic Co^2+^ in the oxidized cobalt film on the silicon wafer [26]. Thus, the XPS spectra results showed that the cobalt was present in the form of Co_3_O_4_, a form containing both Co^2+^ and Co^3+^ species [27]. 

### 4.4. XPS Studies of Reduced Cobalt Metal in Microreactors

The active site during FT synthesis was cobalt in its reduced oxidation state [3]. In general, the reduction behavior of a metal oxide catalyst is studied with the H_2_-temperature-programmed reduction (H_2_-TPR) technique, in which the quantity of hydrogen consumed by the material is evaluated over a range of temperatures [3]. However, the quantification in the case of nanofilms of metal oxide deposited in microchannels is quite challenging. In order to investigate the oxidation state of cobalt catalysts, XPS was performed after the samples were subjected to reduction in the presence of 10%H_2_/Ar at 450 °C for 6 h to evaluate the reducibility of the catalysts. Figure 7 presents the XPS spectra for Co 2p in the reduced cobalt film inside the microchannels of the silicon microreactor. The shift in the Co 2p spectra with no or very low intensity shake-up peaks was compared to the as-deposited cobalt film, confirming that the film was totally reduced to its metal state. The very low intensity of the satellite peaks may be associated with the oxidation losses during the sample transfer post reduction to perform the XPS analysis. The Co 2p spectra for the reduced samples showed two peaks centered at 778.8 eV and 794.3 eV, which correspond to the spin split of Co 2p into Co 2p_3/2_ and Co 2p_1/2_, respectively. The disappearance of satellite peaks indicated that the cobalt oxide film was completely reduced from Co_3_O_4_ to Co, i.e., cobalt in the Co^0^ chemical state on the surface of the microchannels, which served as the active sites during FT synthesis. 

The surface concentration of cobalt present on the outermost layers (2–5 nm) in the microchannels was also investigated in our XPS studies. The composition was calculated from the cobalt 2p peak area to the overall areas. Although the accurate metal content or the quantitative measurement of metal present on the surface could not be determined by XPS, it indicated the amount of cobalt before and after the reduction of the outermost layer of the cobalt film. The surface composition of the cobalt in the oxide state was 18.2% by weight (excluding oxygen), and that of the cobalt in the zero-oxidation state was around 18%. This clearly indicated that there was no significant change in the surface concentration of cobalt after the reduction of the cobalt oxide with hydrogen.

### 4.5. Atmospheric Pressure Fischer–Tropsch Synthesis in Silicon Microchannel Microreactors

The catalytic performance of cobalt film deposited by ALD inside the microchannels of a microreactor was investigated in terms of hydrocarbon (C_1_–C_4_) selectivity, CO conversion, and deactivation studies. Prior to FT synthesis, the catalyst film was activated by its in situ reduction in the microreactor at 350 °C for 6 h, with a constant flow of hydrogen gas (1 sccm). The syngas was fed into the reactor at a constant H_2_/CO molar ratio of 2:1. Nitrogen gas was used as an internal standard, with a continuous flow of 1 sccm at 1 atm throughout the FT studies. The experimental setup for the FT synthesis in the microreactor is shown in Figure 8. For the control experiment, the FT studies were carried out using a bare silicon microreactor, i.e., with no catalyst deposited inside the microchannels at FT operating conditions. The silicon microreactor did not show any CO conversion or hydrocarbon selectivity in the FT operating regime. Therefore, the silicon substrate was considered inert, since it did not show any intrinsic effect on FT synthesis. The reaction attained a steady state within four hours, and the selectivity and CO conversion were calculated at the end of four hours.

Figure 8 shows the bar graph for hydrocarbon selectivity (C_1_–C_4_) at 220 °C and 240 °C. The conversion of CO was observed to be 70% at 240 °C. However, no butane was observed at 240 °C, and the selectivity towards methane, ethane, and propane was 49%, 39%, and 12%, respectively. The selectivity towards methane was 45%, and that of ethane and propane was around 29% and 24%, respectively, at 220 °C. A very small peak for butane was observed, which was approximately 1.8% of the total hydrocarbon selectivity. Since the FT synthesis was carried out at 1 atm, no noticeable products >C_4_ were observed; this is consistent with our previous studies [3,13,28]. Our previous investigation with cobalt loaded into a mesoporous silica-supported catalyst in a 3D-printed microchannel microreactor at 1 atm showed better catalyst activity in terms of CO conversion and hydrocarbon selectivity in the temperature range of 210–240 °C [3]. The performance of cobalt-based catalysts in our previous studies on atmospheric FT synthesis in microchannel microreactors is represented in the Table 1 below. This is also in good agreement with results reported in the literature, showing the best catalyst activity at 220 °C for cobalt-based catalysts in FT synthesis at 1 atm [14,18,29]. 

During the first ~10 h of the reaction, a CO conversion of 75% was observed at 220 °C. While the methane selectivity was higher when compared to the other hydrocarbons at 220 °C at 1 atm, the selectivity to ethane and propane was almost the same, which was 29% and 24%, respectively. The low selectivity towards higher hydrocarbons may be due to FT studies performed at an atmospheric pressure that did not facilitate the polymerization of –(CH_2_) groups on the catalyst surface.

It is interesting to compare the results from our ALD studies to those obtained with sol-gel-coated catalysts. The ALD-deposited cobalt catalysts showed significantly higher selectivity towards C_2_–C_4_, with higher CO conversion over the longer time on stream (TOS) studies [14,17]. While the deposition of cobalt in this work was performed by atomic layer deposition, in which a precursor is pumped into the chamber to react with the hydroxyl group on the silicon microreactor surface, a precursor was hydrolyzed to form a sol-gel for the sol-gel-coated catalysts. This is called one half-reaction, and these half-reactions are responsible for well-controlled film thickness during the ALD reaction cycles [21]. The deposition of cobalt at the atomic scale inside high aspect ratio microchannels facilitates better interaction with reactants. The number of active metal sites in the reaction zone plays a vital role in catalyst performance in terms of CO conversion and selectivity to different hydrocarbons. Previous studies with Fe, Co, and Ru in SiO_2_ sol-gel in silicon microchannel microreactors exhibited a significant effect of metal loading on catalyst reactivity and CO conversion. Ru loading with 4.3% showed the highest selectivity towards higher hydrocarbons, but resulted in very low CO conversion when compared to 10.4% Fe-SiO_2_ and 9.8% Co-SiO_2_ [18] catalysts. The ALD catalyst activity in terms of CO conversion and selectivity towards higher hydrocarbons is explained better in the work performed by Najafabdi et al., where the cobalt catalyst deposited by the ALD technique on porous γ-Al_2_O_3_ was used for FT synthesis at 1 atm, and the FT activity was compared with cobalt impregnated on a similar catalyst support. The results indicated that the ALD catalyst with better dispersion yielded a 38% higher conversion and a 30% higher selectivity towards higher hydrocarbons in CO hydrogenation when compared to the impregnated catalyst at 220 °C [29].

### 4.6. Deactivation Studies of Co-Catalysts Used for Fischer–Tropsch Synthesis

The deactivation studies were carried out at 220 °C at a constant H_2_:CO ratio of 2:1 for 60 h on stream at 1 atm. The ability to withstand deactivation in terms of CO conversion was reported in order to study the long-term catalyst activity. Figure 9 shows CO conversion with respect to time on stream (60 h) at the FT operating conditions mentioned above. The highest CO conversion was observed to be 75% and was constant for the initial 10 h of FT synthesis. A 20% decline in CO conversion was observed after 24 h. The CO conversion remained fairly constant at ~60% after 24 h and was stable until the end of 60 h. The loss of catalyst activity could be explained by different types of deactivation mechanisms, such as the poisoning and sintering of the catalyst, carbon or coke deposition, the negative influence of water vapor during the reaction, and catalyst re-oxidation [2]. It needs to be mentioned that the hydrogen and carbon monoxide, as synthesis gas, were ultrapure research-grade, and the syngas was preheated before sending it into the reactor block; therefore, there was minimal or no possibility of the poisoning of the catalyst that could have arisen from dirty syngas.

The deactivation studies of cobalt catalysts are one of the most significant characteristics of catalyst development and very important for its use in industrial FT synthesis. In order to gain an in-depth understanding of catalyst deactivation during FT synthesis, XPS studies were performed on the spent catalysts after 60 h of time on stream studies. The results were consistent with the long-term stability of the catalyst, as indicated by an increase in the intensity of a weak carbon peak, observed at 284 eV for spent catalyst (Figure 10) when compared to fresh catalyst (Appendix A). This is in contrast with our previous studies with mesoporous silica and mixed oxide composite supports, where mono and bimetallic catalysts deactivated nearly to half of their original activity after 48–60 h [3,4,30]. 

## 5. Conclusions

In summary, the silicon-based microchannel microreactor, designed and microfabricated using conventional microfabrication techniques, is suitable for the conformal deposition of cobalt oxide film inside the microchannels by the ALD technique. The thin film was successfully used as a catalyst for FT synthesis in order to address the significant concerns caused by the conventional coating of the microchannels. As the ALD coating in the microchannels was based on self-limiting chemisorption of the cobalt precursor and water, a conformal catalyst deposition in the high aspect ratio silicon microchannels was obtained. The interactions of cobalt on a silicon wafer in its oxide form and pure metal state were investigated by XPS, making it easier to study the reduction properties of thin film in the microchannels. For atmospheric FT synthesis, the ALD-deposited cobalt-based catalyst showed ~24% propane and 1.8% butane at 220 °C, with a CO conversion of 74%. The deactivation studies showed stronger resistance in terms of CO conversion for 60 h of time on stream during FT synthesis. The fundamental knowledge gained from the FT experiments suggests that ALD is a potential tool for developing novel FT catalysts. 

## Figures and Tables

**Figure 1 nanomaterials-12-02425-f001:**
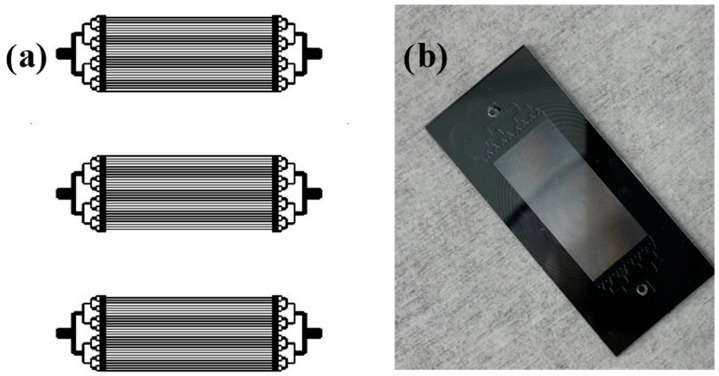
(**a**) AutoCAD design of a photomask and (**b**) an anodically bonded silicon microchannel microreactor.

**Figure 2 nanomaterials-12-02425-f002:**
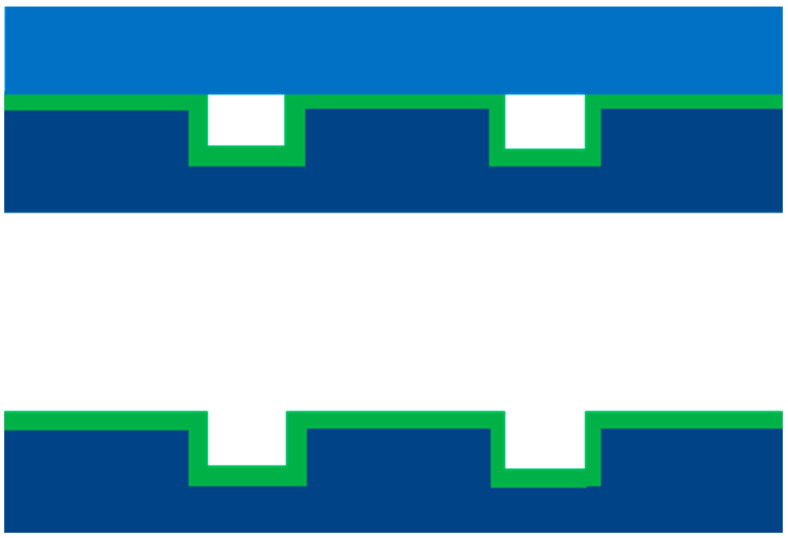
Schematic diagram of ALD-deposited cobalt-based catalyst before and after anodic bonding.

**Figure 3 nanomaterials-12-02425-f003:**
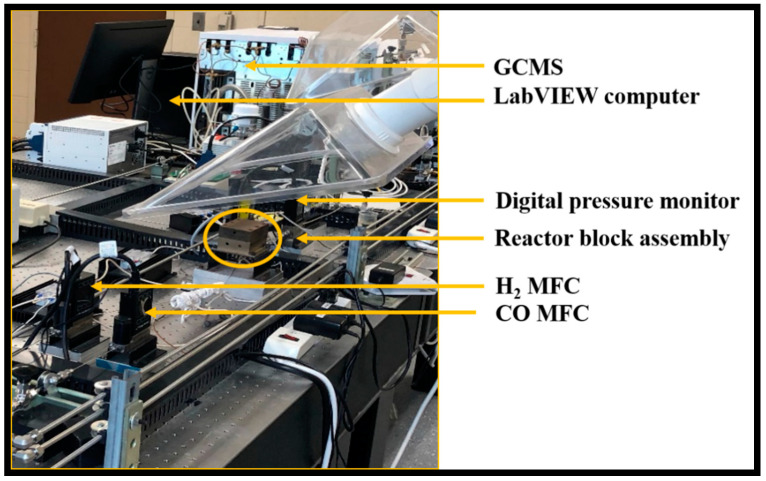
Experimental setup used for Fischer–Tropsch synthesis, using Si-microchannel microreactor [23] (microchannel dimensions: 2.4 cm long × 50 µm wide × 100 µm deep).

**Figure 4 nanomaterials-12-02425-f004:**
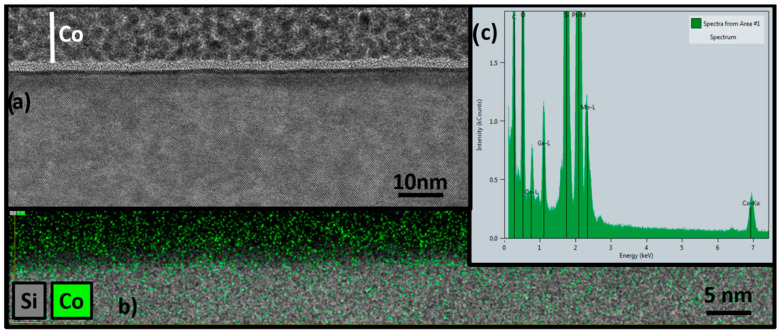
(**a**) Cross-sectional TEM image of cobalt nanofilm on silicon wafer; (**b**,**c**) EDX mapping of cobalt film on the silicon microreactor.

**Figure 5 nanomaterials-12-02425-f005:**
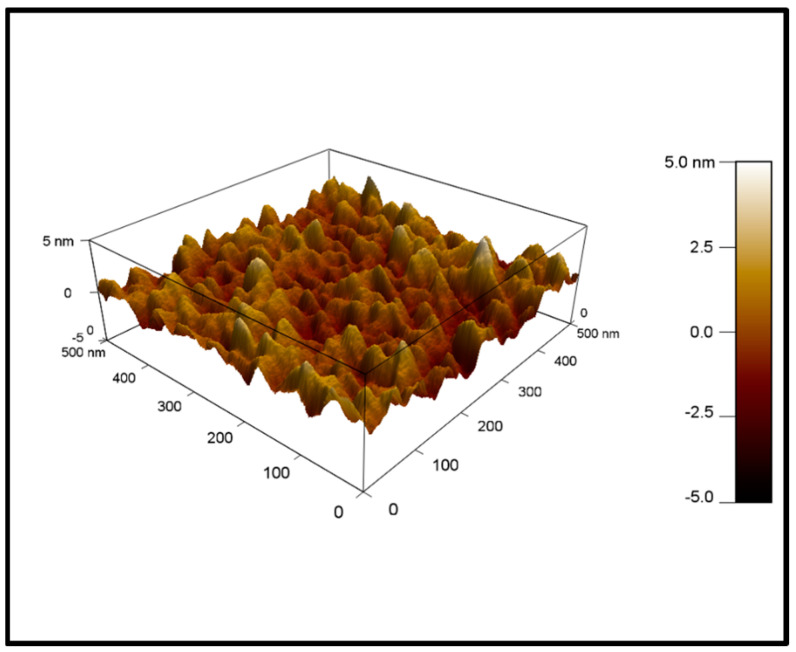
AFM image of cobalt nanofilm deposited on the silicon microreactor.

**Figure 6 nanomaterials-12-02425-f006:**
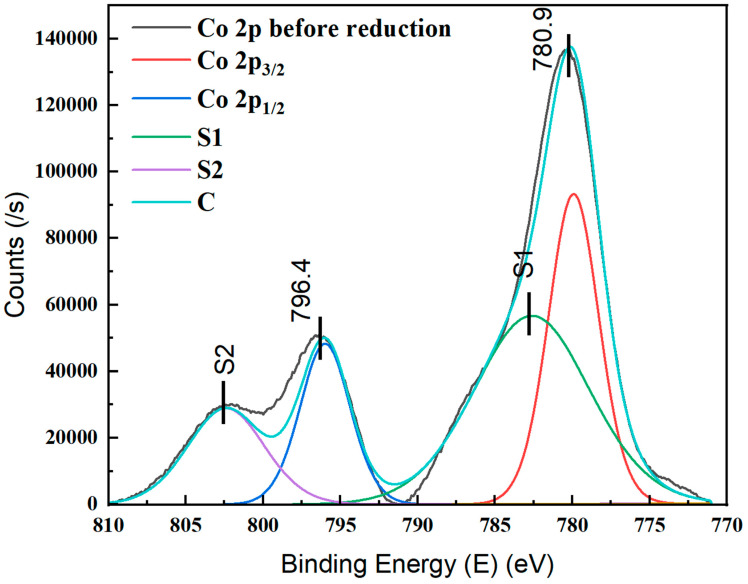
XPS spectra of as-deposited cobalt oxide film by ALD on silicon microreactor.

**Figure 7 nanomaterials-12-02425-f007:**
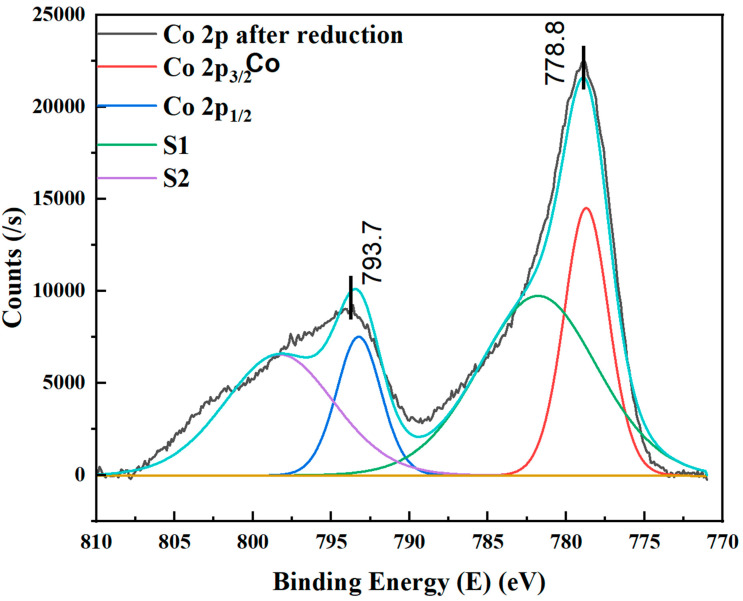
XPS spectra of reduced cobalt film in the microchannels of silicon microreactor.

**Figure 8 nanomaterials-12-02425-f008:**
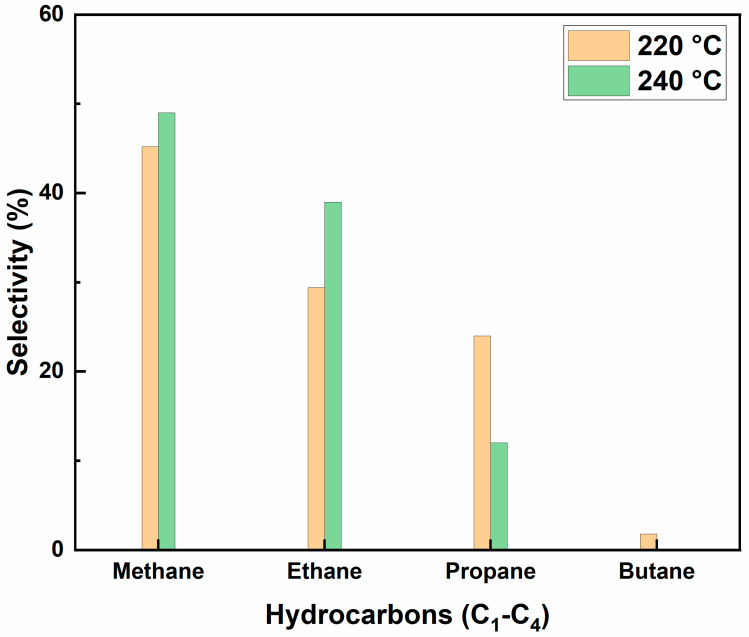
Selectivity of hydrocarbons at 220 °C, with constant H_2_:CO of 2:1.

**Figure 9 nanomaterials-12-02425-f009:**
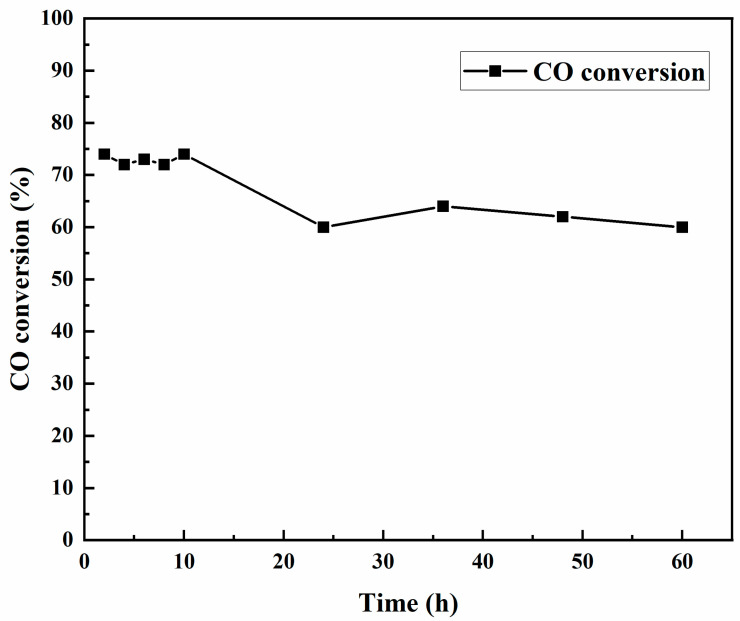
Fischer–Tropsch deactivation studies of cobalt catalyst deposited in silicon microchannel microreactor.

**Figure 10 nanomaterials-12-02425-f010:**
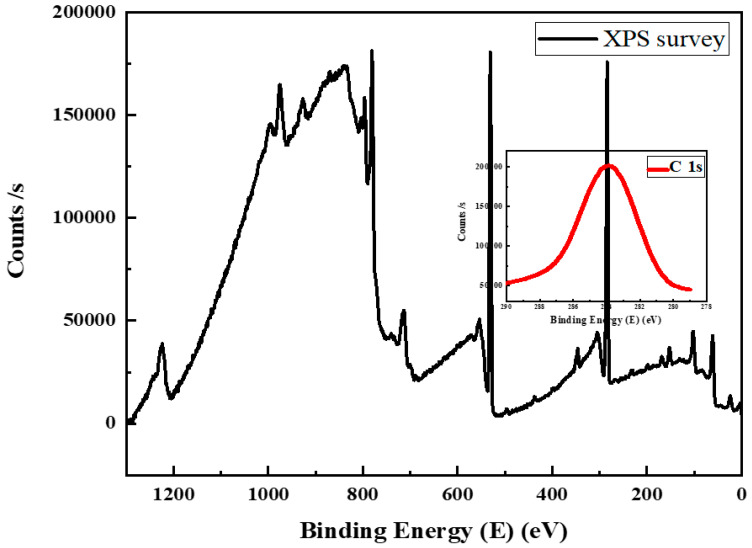
XPS studies of the deactivated catalysts after 60 h of time on stream in silicon microchannel microreactor.

**Table 1 nanomaterials-12-02425-t001:** Performance of cobalt-based catalysts in previous studies on atmospheric FT synthesis in microchannel microreactors.

Catalyst	Type of Reactor	Optimum Temperature	% Co Conversion	Hydrocarbon Selectivity %				Reference
				C_1_	C_2_	C_3_	C_4_	
12% Co TiO_2_	Silicon	250 °C	~95%	95	4	-	-	[14]
12% Co SiO_2_	Silicon	250 °C	~80	45	50	<5	<5	[18]
15% Co MCM-41	3D printed	240 °C	~60%	~75	~15	~6	~4	[3]

## Data Availability

Not applicable.

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
