# Peer review of "Atomic Layer Deposition of Cobalt Catalyst for Fischer–Tropsch Synthesis in Silicon Microchannel Microreactor"

_nanomaterials, 2022, doi:10.3390/nano12142425_

Round 1

Reviewer 1 Report

The article is devoted to the study of the properties of cobalt-based nanofilms obtained by the atomic layer deposition method, as well as their use as catalysts for atmospheric Fischer-Tropsch synthesis. Undoubtedly, the results presented by the authors are of high scientific novelty and practical significance, and are also promising for practical research. In general, the presented results of the study can be accepted for publication after the authors provide answers to all the questions raised by the reviewer during the reading of the article.

1. In the abstract, the authors need to more clearly state the purpose and relevance of this work.
2. Authors should change the presentation of the results, some of them can be combined together or put into an appendix.
3. The authors should clarify whether the structural parameters of the films, in particular the degree of crystallinity, have been determined.
4. How exactly the thickness of the obtained films was determined, only according to microscopy data or even with the use of any methods.
5. Authors should compare their results with other types of catalysts.
6. Conclusion requires significant revision.

Author Response

Atomic Layer Deposition of Cobalt Catalyst for Fischer-Tropsch Synthesis in Silicon Microchannel Microreactor

Nafeezuddin Mohammada, Shyam Aravamudhana, Debasish Kuilaa,b,*

a Department of Nanoengineering, Joint School of Nanoscience and Nanoengineering

b Department of Chemistry

North Carolina A&T State University, Greensboro, NC 27411, USA

Dear Editor,

Ref - Manuscript number: nanomaterials-1688332

The authors would like to thank the editor and reviewers for a thorough review of our manuscript- ‘Atomic Layer Deposition of Cobalt Catalyst for Fischer-Tropsch Synthesis in a Silicon Microchannel Microreactor’ submitted to Nanomaterials. In the revised manuscript, we have addressed the reviewer’s comments and recommendations.   All new insertions/additions are highlighted in bold in the revised manuscript as well as in the rebuttal. We sincerely believe that this in-depth review has significantly improved the quality of our manuscript.

Thank you for consideration of our manuscript for publication

Sincerely

Debasish Kuila, Nafeezuddin Mohammad

Reviewer's comments:

1. In the abstract, the authors need to more clearly state the purpose and relevance of this work.

Thank you for your thorough review. We have made multiple changes in the abstract and a part of the discussion for characterization and results of the manuscript considering the reviewer’s comments.

  1. Authors should change the presentation of the results, some of them can be combined together or put into an appendix.

Thank you for your suggestion. We have made some modifications in the presentation of results and described our results in a very detailed way for an easy read for a broad range of audiences.  

  1. The authors should clarify whether the structural parameters of the films, in particular the degree of crystallinity, have been determined.

The present scope of this research study is to synthesize some innovative catalysts using the ALD technique for robust miniaturized microchannel microreactors. We have successfully deposited the cobalt oxide film and performed an in situ reduction of cobalt oxide to reactive cobalt within the microchannels for Fischer-Tropsch Synthesis.

Crystallinity studies: An attempt was made to study the effect of crystallinity of the cobalt film on the performance of the catalyst before submitting the manuscript; however,  in this case, the catalyst is a cobalt nanofilm that is deposited on the surface of the microchannels of microreactor which makes challenging to study the crystallinity effect on the reactor parameters and catalyst performance. However, in our powder-based catalysts, we have studied the effect of the support structure and the crystallinity on FT synthesis.

Abrokwah, R. Y., Rahman, M. M., Deshmane, V. G., & Kuila, D. (2019). Effect of titania support on Fischer-Tropsch synthesis using cobalt, iron, and ruthenium catalysts in silicon-microchannel microreactor. Molecular Catalysis478, 110566.

Mohammad, N., Abrokwah, R. Y., Stevens-Boyd, R. G., Aravamudhan, S., & Kuila, D. (2020). Fischer-Tropsch studies in a 3D-printed stainless steel microchannel microreactor coated with cobalt-based bimetallic-MCM-41 catalysts. Catalysis Today358, 303-315.

  1. How exactly the thickness of the obtained films was determined, only according to microscopy data or even with the use of any methods.

The thickness of cobalt nanofilm is determined by cross-sectional TEM image as well as ellipsometry. The Savannah ALD instrument which is used to deposit the cobalt oxide in this case is equipped with in-situ ellipsometry that monitors thickness with the deposition process time. The final thickness of the cobalt nanofilm is almost equal to the thickness determined by the TEM image.

  1. Authors should compare their results with other types of catalysts.

We thank the reviewer for suggesting incorporating the comparison results for the FT synthesis in the microreactor. We have included a table that presents the performance of different types of catalysts for the FT synthesis in microchannel microreactor (both silicon and 3D printed microchannel microreactor.

  1. Conclusion requires significant revision.

We have made significant changes to the content of the manuscript in the Abstract, results and discussion as well as the conclusion part of the revision.

Reviewer 2 Report

The authors reported a fabrication of Co catalyst in the microchannel reactor with ALD technique, C1~C4 products were observed at high conversion under atmospheric pressure. The work provides a possibility to prepare a Co-based catalyst with the ALD technique inside the Si-microchannels. The results are interesting for publication, however, several points need clarifying and require further justification. These are given below.

  1. The fitting of the XPS spectra does not fit well with the original one, please double-check the result(figure 6,7). For instance, the fitting of the satellite peak of Co 2p 3/2 was not included.

In fig. 10, The XPS survey spectrum of the fresh catalyst should be included and compared with the deactivated catalyst.

  1. How about the carbon balance in the F-T synthesis?
  2. The influence of the flow rate toward selectivity was not discussed/provided.
  3. The flow rate was provided, would be better to provide the microchannel volume, so that reader can estimate the residence time.
  4. Since the catalyst was fabricated inside the microreactor, it is important to evaluate the possibility to regenerate the catalyst. It is recommended to test the catalyst activity after regeneration.

Author Response

  1. The fitting of the XPS spectra does not fit well with the original one, please double-check the result(figure 6,7). For instance, the fitting of the satellite peak of Co 2p 3/2 was not included.

We have performed XPS fitting using origin software to get a reasonably high-resolution core XPS spectra to separate the signal of elemental oxidation states. Figures 6 and 7 show the spectra for the cobalt oxide as-deposited and cobalt metal after the reduction. The spectra clearly show the difference in the oxidation states by having an additional satellite peak for the cobalt oxide which confirms that the reduction under hydrogen has generated a reactive/active metal catalyst on the outermost layer of the film/surface of the microchannels.

  1. In fig. 10, The XPS survey spectrum of the fresh catalyst should be included and compared with the deactivated catalyst.

Thank you for the suggestion, we have included the XPS spectrum of the fresh catalysts in the supplementary part of the manuscript.

  1. How about the carbon balance in the F-T synthesis?

We have not reported the carbon balance for the Fischer-Tropsch synthesis because of a very small amount of catalyst (almost in nano/picograms) which is active for FT synthesis in microchannels of the microreactor. The deposition of carbon over the time on stream will have a significant effect on the overall carbon balance. However, this is not the scope of the current research studies, the performance of the catalyst was evaluated in terms of CO conversion as well as the selectivity towards hydrocarbon.

  1. The influence of the flow rate toward selectivity was not discussed/provided.

The initial studies for the FT synthesis in the microchannel microreactor were performed by varying the temperature and flow rates. The optimized flow rates in terms of selectivity towards hydrocarbon were selected to compare the outcome of FT synthesis at two different temperatures. The overall objective of the study is to validate the cobalt nanocatalyst as a prominent FT synthesis, therefore the optimization of the reaction performance is not reported in the current research study.

  1. The flow rate was provided, would be better to provide the microchannel volume, so that reader can estimate the residence time.

The number of microchannels, as well as the dimensions of microchannels fabrication on the silicon wafers, is detailed in the experimental fabrication steps in the manuscript. We have included the volume of all the microchannels in the results and discussion section.

  1. Since the catalyst was fabricated inside the microreactor, it is important to evaluate the possibility to regenerate the catalyst. It is recommended to test the catalyst activity after regeneration.

The reviewer suggested evaluating the possibility of regeneration of catalyst after performing FT synthesis. We have conducted the deactivation studies of the cobalt nanofilm which is deposited on the microchannels of the microreactor at 220 °C only. The FT synthesis was performed over the 40 hours on stream at 220 °C only to evaluate the long-term stability of the cobalt catalyst when compared to other powdered/slurry-based catalysts used for FT synthesis in the microchannel microreactor. Our previous studies indicate that the catalyst deactivated much faster when compared to the current ALD deposited cobalt nanofilm. Also, the deactivation studies were carried out only for the 40 hours and the catalyst was characterized after that to get more insights into the deactivation as well as to have a better comparison with powdered/sol gel based catalysts.

Reviewer 3 Report

The manuscript titled “Atomic Layer Deposition of Cobalt Catalyst for Fischer-Tropsch Synthesis in Silicon Microchannel Microreactor” presented by Nafeezuddin Mohammad, Shyam Aravamudhan, Debasish Kuila deals with the microchannel reactor where the active component is prepared by ALD. Although the ALD is expensive method for synthesis of catalyst it could be applied to investigate the model reactor and model reactions. The manuscript could be interested to wide group of researchers. However, there are major issues that could be clarified:

  • The abstract is absent in the manuscript file.
  • Lines 69-70, 118: The authors use the abbreviations without full name all of them should be written in the full name when they are firstly introduced.
  • The authors used the imperial and metric units.
  • Section 2.3 should be extended, and the ALD process should be explained mor carefully as well as the description of Figure 2.
  • Section 2.4: The information about mass flow controllers is absent.
  • Section 3. The information about TEM and AFS microscope is absent (model, equipment, operating voltage, resolution, etc.). The XPS setup is poorly described (analyzer, X-ray source). The description of XPS analysis is poor (background, lineshapes, charge correction, etc.).
  • As I understand correctly the shape of reactor is complicate, and cobalt oxide film does not be studied by TEM, XPS, AFM directly without a special preparation for study. The information about the preparation procedure is demanded.
  • Lines 135-136: Authors state that “The image indicates the presence of cobalt oxide as a polygranular film on 135 the silicon wafer and it is uniform throughout the sample”. That statement is doubt and should be proved by TEM study of a few points along the microchannel. For example, in the beginning, in the center and at the end of microchannel.
  • Figure 4c is clearly demonstrate of the presence of Pt, Mo, and Ga. Please, could the authors discuss this observation?
  • Lines 146-147: “significantly less roughness” - comparison with what?
  • The sections (4.3 and 4.4) devoted to XPS analysis is badly written. The authors made a few gross mistakes: incorrect fitting, use of incorrect special terms, corresponding the binding energies to the wrong chemical states. I strongly recommend the authors to see how it is presented by the other scientists and then carefully rewrite the XPS section.
  • Line 192: In situ.
  • Line 194: Figure 8 does not demonstrate the “experimental setup”.

The choice of methods to study the objects is adequate, but the obtained results demand the careful explanation and discussion. The manuscript demands the strong proof readings.

Author Response

  • The abstract is absent in the manuscript file.

The abstract is now added to the revised manuscript as well as the current editorial version.

  • Lines 69-70, 118: The authors use the abbreviations without full name all of them should be written in the full name when they are firstly introduced.

Thank you for pointing it out, we have incorporated the full names of those abbreviations while referring for the first time in the revised manuscript.

  • The authors used the imperial and metric units.

All the dimensions are presented as per the metric units.

  • Section 2.3 should be extended, and the ALD process should be explained more carefully as well as the description of Figure 2.

ALD section has been extended in the revised manuscript.

  • Section 2.4: The information about mass flow controllers is absent.

Included in the revised version

  • Section 3. The information about TEM and AFS microscope is absent (model, equipment, operating voltage, resolution, etc.). The XPS setup is poorly described (analyzer, X-ray source). The description of XPS analysis is poor (background, lineshapes, charge correction, etc.).

The information on characterization tools has been updated and detailed as much as possible

  • As I understand correctly the shape of reactor is complicate, and cobalt oxide film does not be studied by TEM, XPS, AFM directly without a special preparation for study. The information about the preparation procedure is demanded.

No special preparation for the sample is needed, the characterizations were performed as per the regular thin film characterization on the bare silicon wafer shard.

  • Lines 135-136: Authors state that “The image indicates the presence of cobalt oxide as a polygranular film on 135 the silicon wafer and it is uniform throughout the sample”. That statement is doubt and should be proved by TEM study of a few points along the microchannel. For example, in the beginning, in the center and at the end of microchannel.

The granular morphology is observed in AFM which is also clearly seen in the TEM image, this is reported already in the manuscript.

  • Figure 4c is clearly demonstrate of the presence of Pt, Mo, and Ga. Please, could the authors discuss this observation?

This is because the sample needs a thin coating og those metals for charging while imaging and EDX. The overall objective of the TEM characterization in this study is to study morphology, calculate the thickness as well as confirm the deposition of copper via ALD technique

  • Lines 146-147: “significantly less roughness” - comparison with what?

When compared to the other powdered catalyst which was coated by sol-gel techniques. This sentence is improved in the revised version of the manuscript

  • The sections (4.3 and 4.4) devoted to XPS analysis is badly written. The authors made a few gross mistakes: incorrect fitting, use of incorrect special terms, corresponding the binding energies to the wrong chemical states. I strongly recommend the authors to see how it is presented by the other scientists and then carefully rewrite the XPS section.

We have performed XPS fitting using origin software to get a reasonably a good resolution core XPS spectra to separate the signal of elemental oxidation states. Figures 6 and 7 show the spectra for the cobalt oxide as-deposited and cobalt metal after the reduction. The spectra clearly show the difference in the oxidation states by having an additional satellite peak for the cobalt oxide which confirms that the reduction under hydrogen has generated a reactive/active metal catalyst on the outermost layer of the film/surface of the microchannels. This analysis is based on our previous published XPS studies as well as referred to and by many other researchers in this area. Also, some discussion on the XPS is added.

  • Line 192: In situ.

This is the correct terminology.

  • Line 194: Figure 8 does not demonstrate the “experimental setup”.

This is updated.

Round 2

Reviewer 1 Report

The authors answered all the questions of the reviewer, the article can be accepted for publication.

Author Response

Thank you so much for the review. 

Reviewer 2 Report

I still have doubts with the xps peak fitting and discussion, in my opinion, satellite peaks still present after reduction.

Author Response

The cobalt oxide was reduced and the sample is transferred for XPS characterization in a sampling jar, the presence of a very lite satellite could be a result of oxidation losses during the transferring process. The complete reduction of metal oxide to metal is quite difficult unless there is an insitu reduction of metal oxide film with in the XPS chamber

Reviewer 3 Report

Authors have improvrd the manuscript. They answered all comments, but there are few issues:

1. The use of original software do not guarantee the correct fitting. Taking into account the ratio between Co2p3/2 and Co2p1/2 to 3:2 one can see that the both component have the similar integral intensities. Moreover, the FWHM of Co2p1/2 is too big, it should be slightle wider than Co2p3/2 (+0.2...0.5 eV). Secondly, the authors introdused into fitting model the satellite peak for Co2p1/2 component (Fig. 6), but did not do the same for Co2p3/2 component. Thus, I should say that despite the experience of the authors this fitting could not be called correct.

2. I recommend authors to merge both XPS spectra into one figure to clearly demonstrate the observed changes.

Author Response

Thank you very much for the suggestion, we have updated the XPS fitting using avantage software for better understanding of surface properties of the film and added it to the revised manuscript.